# Simultaneous Quantification of Seven Antifungal Agents in Human Serum Using Liquid Chromatography-Tandem Mass Spectrometry

**DOI:** 10.3390/ph16111537

**Published:** 2023-10-30

**Authors:** Wenjing Li, Yang Li, Junlong Cai, Yue Wang, Yanan Liu, Hankun Hu, Liang Liu

**Affiliations:** 1Department of Pharmacy, Zhongnan Hospital of Wuhan University, 169 Donghu Road, Wuhan 430071, China; liwenjing@znhospital.cn (W.L.); wyddzyx2008@163.cm (Y.W.); liuyanan1037@163.com (Y.L.); huhankun@whu.edu.cn (H.H.); 2Department of Blood Transfusion, Zhongnan Hospital of Wuhan University, Wuhan 430071, China; liyang19890115@126.com; 3Department of Clinical Trial Center, Zhongnan Hospital of Wuhan University, Wuhan 430071, China; zn002661@whu.edu.cn

**Keywords:** antifungal agents, therapeutic drug monitoring, LC–MS/MS, simultaneous quantitation, human serum

## Abstract

Systemic antifungal agents are essential for high-risk patients undergoing immunosuppressive therapy or cancer chemotherapy because of the rapid increase in opportunistic fungal infections. Therapeutic drug monitoring is crucial to ensuring the efficacy and safety of antifungal agents owing to their pharmacokinetic variability. In the present study, we developed and validated a quantitative method for the simultaneous detection of seven commonly used antifungal drugs (amphotericin B, isavuconazole, voriconazole, fluconazole, posaconazole, caspofungin, and micafungin) using liquid chromatography-tandem mass spectrometry. Methanol (containing 0.1% formic acid) was used for protein precipitation and only 50 μL of serum was required for the analysis. Chromatographic separation was conducted using a Waters Acquity UPLC C8 column, and one stable isotope-labeled agent and two analogs were used as internal standards. The calibration curves ranged from 0.1 to 50 μg/mL for all agents, and the correlation coefficient (R^2^) for all calibration curves was above 0.9835. The intra-day precision (1.2–11.2%), inter-day precision (2.4–13.2%), and mean bias values (−10.9 to 13.6%) were within an acceptable range of ±15%. Successful implementation of the developed method in clinical practice would facilitate the effective monitoring of these antifungal agents.

## 1. Introduction

Invasive fungal diseases remain a prevalent cause of mortality in immunocompromised patients [1,2,3]. This challenge is attributed primarily to the scarcity of reliable and secure antifungal therapies, particularly for patients undergoing transplantation, transfusion, and intensive care [4]. A prospective study revealed that *Candida* and *Aspergillus* infections accounted for 17% and 1.4%, respectively, of infection cases among intensive care unit patients [5]. The increasing incidence, mortality, and escalating issue of drug resistance associated with invasive fungal infections highlight the necessity of administering antifungal drugs with a focus on early, long-term, and rational dosing [4]. When using antifungal agents, clinicians meticulously evaluate their antifungal spectrum, therapeutic effect, tolerability, and potential toxicity; however, the current standard dosing routines for antifungal drugs are often prescribed based solely on a patient’s body weight without considering other essential factors [6,7]. This may lead to inadequate or excessive drug concentrations in the blood, potentially contributing to unfavorable clinical outcomes. Patients with invasive fungal infections often undergo immunosuppressive therapy, transplantation, cancer chemotherapy, or treatment for multiple bacterial infections, and thus experience significant physiological changes such as liver damage, kidney impairment, extravascular fluid loss, and sepsis-related inflammation. These complications can affect the pharmacokinetics (PK) of antifungal drugs, resulting in heterogeneous PK parameters [8]. Therapeutic drug monitoring (TDM) holds considerable potential to enhance the treatment of invasive fungal infections by adjusting the dosage of an antifungal agent based on its PK in an individual patient [9,10]. A meta-analysis of ten randomized controlled trials revealed that individualized antimicrobial dose optimization resulted in a significantly higher rate of therapeutic goal achievement, reduced treatment failure, and reduced renal toxicity [11].

Considering the significant morbidity and mortality associated with invasive fungal infections, combination therapy involving multiple antifungal agents may be an option, particularly in cases of severe infection [12,13]. Recent clinical data on *cryptococcosis* have shown that a combination of amphotericin B and flucytosine can be used to effectively treat patients with *cryptococcal* meningitis [14]. A recent clinical trial on *candidemia* also indicated that the combined use of amphotericin B with fluconazole could yield better clinical outcomes than using fluconazole alone [15]. Although combination antifungal therapies can improve the efficacy of fungicides, drug combinations can also affect the pharmacokinetics/pharmacodynamics (PK/PD) of the drugs through complex mechanisms. The combined application of antifungal drugs has a reciprocal impact, with azoles reducing the activity of amphotericin B by consuming or changing the ergosterol target [13]. Therefore, it is essential to develop a method capable of simultaneously measuring the levels of multiple antifungal agents, expediting clinical response time, and optimizing the allocation of manpower and material resources. 

Various methods have been developed to measure the levels of antifungal agents in serum samples. Immunoassays are a common method that are readily implemented in routine laboratory procedures [16,17]; however, immunoassays have certain limitations, such as low specificity and accuracy and the ability to analyze only a limited number of drugs [18,19]. Several high-performance liquid chromatography-ultraviolet (HPLC–UV) detection methods have been reported for determining antifungal drug concentrations in sera; however, these methods typically require large sample sizes and cannot analyze multiple drugs simultaneously [20,21]. In contrast, liquid chromatography-tandem mass spectrometry (LC–MS/MS) has the advantages of a small sample size requirement, streamlined sample preparation, high sensitivity, good reproducibility, and the ability to handle multiple analytes [22,23].

In this study, we developed an LC–MS/MS-based platform for monitoring seven antifungal agents, namely, polyenes (amphotericin B), triazoles (fluconazole, voriconazole, posaconazole, and isavuconazole), and echinocandins (caspofungin and micafungin) in human serum. These antifungal agents are commonly used to treat invasive fungal infections. TDM of specific agents, such as fluconazole and voriconazole, is routinely recommended, whereas for other agents, TDM is recommended only for certain patients [1,24]. We found that the developed method is accurate, simple, rapid, and economical, and requires a minimal serum volume of 50 μL. Therefore, this method can potentially be used for monitoring routine serum drug concentrations in clinical practice to enhance therapeutic effects.

## 2. Results and Discussion

### 2.1. Method Establishment

In the first stage of the study, pure chemical standards were employed to develop and optimize the mass spectrometry method. A solution with a concentration of 5 μg/mL was used for this purpose. The method development process was initiated by refining the ionization and fragmentation conditions for the analyte and internal standard (IS). Typically, the precursor ion *m*/*z* was obtained by adding or subtracting a proton from the molecular mass of the original molecule, resulting in ions such as [M + H]^+^ for positive ionization, or [M − H]^−^ for its negative mode. For example, the precursor ions of voriconazole and isavuconazole are simply charged ions with *m*/*z* values of 350.2 and 717.2, respectively. In some cases, the precursor ion may be doubly charged, in which case it would be represented as [M + 2H]^2+^/2 or [M − 2H]^2−^/2. In this study, only the precursor ion of caspofungin is a doubly charged ion with an *m*/*z* value of 547.40. Specifically, the precursor ion of amphotericin B is 906.5; however, its molecular weight is 924.5, which may be due to the loss of one molecule of water in the molecule. Following the acquisition of precursor ions, the product ion mode was employed to screen and select suitable product ions. Three or four product ions were selected after optimizing the collision energy (CE). The precursor and product ions were combined to form multiple reaction monitoring (MRM) transitions for each compound, and the transitions were further validated by spiking the blank serum extraction solution with pure compounds. Transitions with the highest signal-to-noise ratio (S/N) were selected for quantitative analysis. The parent and product ions used in these analyses are listed in Table 1. For quantification and confirmation, two distinct *m*/*z* transitions were selected for each analyte [25]. All analytes were detected in the positive mode of electrospray ionization (ESI), except for micafungin and ethylparaben (used as the internal standard for micafungin), which exhibited insufficient intense ion fragmentation in this mode. Conversely, the response intensity was excellent in the negative ion mode, thus, it was detected in the negative mode. To ensure accurate quantification, it is recommended to have a minimum of 10 scans per peak for the dwell time during the cycle time. In our study, we set the dwell time for each MRM transition at 20 msec, and smooth peak shapes were obtained.

Subsequently, three columns were tested to determine the most effective separation, including SHIMADZU C8 (2 μm, 2.1 mm × 100 mm), Waters Acquity UPLC C8 (1.7 μm, 2.1 mm × 50 mm), and Agilent SB C18 (2.7 μm, 2.1 mm × 30 mm). The Waters Acquity UPLC C8 column was selected because of its exceptional ability to achieve superior chromatographic separation. An ammonium acetate solution (2 or 10 mM) was incorporated into the mobile phase to enhance peak quality and detect pH-sensitive compounds; however, ammonium acetate addition did not significantly enhance the peak characteristics compared with the results obtained in its absence. The analytes are not chromatographically resolved but that is not needed in LC–MS/MS, which has the advantage of introducing a new dimension in the separation. Indeed, it is important that these peaks are monitored in different transitions, and our results indicate that their co-elution does not interfere with the quantitative analysis. 

After selecting the column and mobile phase, the mobile phase gradient was varied to optimize the peak shape and resolution. During the initial stage of this research, the profile of the first gradient tested was as follows: phase B started at 10% and remained constant for 1 min, then increased to 90% in 4 min and remained constant for 4 min, and finally decreased to 10% in 1 min. However, the retention times of isavuconazole were extremely short. After testing different gradients, the gradient of B was finally set as 5% (0 min), 50% (3 min), 100% (4 min), 100% (7 min), 5% (9 min), and 5% (10 min). This gradient program successfully resulted in well-resolved peaks, facilitating precise identification and quantification of the target analytes. A representative chromatogram is shown in Figure 1. The presence of an undesired tailing in the peak shape of fluconazole may be attributed to a combination of interactions with the stationary phase and matrix effects. 

Subsequently, sample pretreatment was optimized. Different organic solvents, including methanol, methanol/acetonitrile (50:50), pure acetonitrile, methanol with 0.1% formic acid (FA), methanol/acetonitrile (50:50, 0.1% FA), and acetonitrile with 0.1% FA, were tested to optimize precipitation efficiency. The peak areas of the samples that spiked after protein precipitation were compared with those of standard solutions at equivalent concentrations to evaluate the analyte recovery. Among the various solvents tested, methanol with 0.1% FA provided the highest and most consistent recovery rate. 

Owing to the complexity of biological samples, the matrix often significantly interferes with bioanalytical methods, particularly when ESI mass spectrometry is used [26]. Selecting different ISs is justified owing to their ability to correct and replicate the analytical behavior of each agent. We used VOR-IS ([^13^C_2_,^2^H_3_]-voriconazole), tylosin, and ethylparaben as ISs, considering a balance between cost and accuracy. In the subsequent analysis, all three ISs demonstrated excellent measurability and effectively corrected the standard curves.

### 2.2. Method Validation

#### 2.2.1. Linearity, Precision, and Accuracy

The key to ensuring assay performance and data quality is to establish and interpret a good calibration curve, particularly in bioanalytical LC–MS/MS assays used in clinical practice. In this study, the relative peak ratios of the analyte and ISs were plotted on the y-axis to construct a calibration curve. The least-squares regression method was used with a linear regression weighting factor of 1/x^2^ [27]. As shown in Figure 2, the representative calibration curves displayed sufficient linearity for quantifying all antifungal agents, with the correlation coefficients (R^2^) being above 0.9835. The concentration range of the antifungal agents was carefully selected as 0.1–50 μg/mL to ensure correspondence with their actual concentrations in clinical blood samples and to avoid signal saturation issues caused by excessively high concentrations. Following the Food and Drug Administration (FDA) guidelines on bioanalytical method validation, we confirmed that the target analyte/blank matrix analytical signal ratio exceeded five in the target analyte retention time window. However, the limit of quantification (LOQ) was deliberately set well below the effective concentration range required for clinical TDM. In this study, the lower LOQ (LLOQ) was set at 0.1 μg/mL for all antifungal agents; this is significantly higher than the LOQ recommended by the FDA.

The accuracy and precision of the assays were evaluated at four quality control (QC) levels: LLOQ, low QC (LQC), medium (MQC), and high QC (HQC). The inter- and intra-day precision values are summarized in Table 2. The precision for the intra-day LLOQ ranged from 4.4% to 11.9%, and the inter-day precision ranged from 4.1% to 13.2%. Correspondingly, the accuracy bias values ranged from 2.8% to 16.1%. All these parameters meet the FDA’s LLOQ requirements. The intra-day precision for the remaining QC samples ranged from 1.2% to 11.2%, and the inter-day precision ranged from 2.4% to 13.2%. The accuracy bias value ranged from −10.9% to 13.6%. All values obtained are within the recommended range of ±15% for precision and accuracy, as specified by the FDA.

#### 2.2.2. Specificity and Selectivity

Representative chromatograms of the matrix blank and LLOQ samples are shown in Figure 3. Neither the spiked serum samples from healthy volunteers nor those obtained from patients with hemolysis, icterus, or lipemia (HIL) demonstrated any interference at the specified retention time windows for each target MRM. Furthermore, it should be noted that isavuconazole, ethylparaben, and tylosin may not be completely separated in the chromatography; however, they do not interfere with each other in their respective transitions.

#### 2.2.3. Extraction Recovery and Matrix Effect

The mean extraction recoveries from the QC samples at low and high levels (three-fold LLOQ and 80% upper limit of quantification (ULOQ)) are summarized in Table 3. All antifungal agents exhibited excellent recoveries in the range of 82.27–105.24%, with relative standard deviations (RSDs) of 3.6–9.5%. Notably, no significant difference was observed between spiked serum samples from healthy volunteers and those from patients with HIL. In ESI-MS, the matrix effect is crucial when measuring several substances in a single bioanalytical test. In our study, the mean matrix factors ranged from 1.17 to 2.21 at the LQC and HQC levels. The RSDs were less than 15% for all analytes, indicating the robust technical reproducibility of the assay.

#### 2.2.4. Stability

The stability of analytes is an essential factor in clinical testing, particularly under given temperature conditions. The mean stability values and standard deviations of LQC and HQC samples are summarized in Table 4 (*n* = 6). The minor differences were within the clinically acceptable range, indicating the satisfactory stability of the analytes.

### 2.3. HPLC Validation

HPLC analysis of voriconazole was performed for external quality assessment during the method validation process. HPLC was performed as described by Miao et al. [28]. A total of 33 samples were analyzed using both methods for comparison. Passing–Bablok regression analysis was conducted using MedCalc software (V. 20.014), which confirmed a strong correlation between the LC–MS/MS and HPLC assays (y = −0.115 + 1.137x), as depicted in Figure 4A. Additionally, a Bland–Altman plot analysis revealed no significant linear differences between the two methods, as shown in Figure 4B.

### 2.4. Application

The developed method was used for TDM in 48 patients who underwent antifungal drug therapy during hospitalization. A total of 104 serum samples were collected immediately before the administration of antifungal drugs (within half an hour) and after the completion of the fifth to seventh administration. The mean (standard deviation) serum concentrations of these agents are shown in Figure 5. Although the recommended range is 1.0–5.0 μg/mL [29], the average voriconazole level was 3.2 μg/mL, ranging from 0.12 to 9.33 μg/mL. These results signify that 30.30% (10/33) of the patients did not fall within the recommended range. The quantification of drug concentrations provides a valuable reference for assessing changes in drug disposition and guiding dose adjustments in conjunction with clinical evaluation. Specifically, voriconazole metabolism exhibits nonlinear PK characteristics with significant inter-individual variability, highlighting the importance of drug concentration adjustments for personalized medicine dosage. The established method for the antifungal agent TDM contributes to the safe and effective administration of these medications.

For instance, a 61-year-old male patient was admitted to the ICU with a severe lung infection caused by drug-resistant *Acinetobacter baumannii* and *Aspergillus fumigatus*. The treatment regimen included 0.6 g of linezolid and 500,000 U of polymyxine every 12 h, 3 g of cefoperazone sulbactam every 8 h, an initial dose of 0.5 g of voriconazole every 12 h, and voriconazole was maintained with a dose of 0.3 g every 12 h after the initial loading dose. After four days, liver function test results revealed significantly elevated levels of alanine aminotransferase, glutamate aminotransferase, and indirect bilirubin, which were more than 16 times higher than the upper limit of the normal reference range. Blood analysis revealed a voriconazole concentration of 9.96 μg/mL, which exceeded the recommended range of 5.5 μg/mL. This result suggests that voriconazole may cause acute liver injury; therefore, the dose was adjusted to 0.2 g every 12 h. After eight days, the voriconazole concentration decreased to 4.56 μg/mL, which was within the recommended range. After two weeks, liver enzyme and bilirubin levels returned to normal.

### 2.5. Comparison with Reported Methods

Multiple factors, such as the chemical properties of a drug, metabolic pathways, required precision for clinical use, specimen volume and throughput, and testing costs, should be considered when selecting an appropriate analytical platform. Immunoassays offer the advantage of being easily performed on automated, high-speed clinical analyzers typically found in most hospital laboratories; however, their usefulness is affected by some issues: their scarce specificity and accuracy. HPLC methods have been widely reported for the determination of antifungal agents in serum and are still used in major reference laboratories as these techniques demonstrate excellent reproducibility and stability; however, they often require complex extraction procedures (such as solid-phase extraction or liquid–liquid extraction), large sample sizes, and time-consuming chromatographic steps and are not capable of analyzing multiple drugs simultaneously. Notable advantages of mass spectrometry are its compatibility and multiplexing ability, which allow the concurrent evaluation of a variety of substances. Furthermore, LC-MS/MS is more sensitive than conventional HPLC. Analyzing multiple drugs simultaneously using a single platform can be more convenient and time-efficient, particularly when dealing with patients who require multiple antifungal agents.

The LC–MS/MS approach developed and validated in this study enables the simultaneous determination of seven antifungal drugs. It requires only a small serum volume of 50 µL and has a chromatographic run time of 10 min. Recently, other procedures have been reported for the simultaneous determination of antifungals, as shown in Table 5. Yi et al., reported a method for detecting six compounds including fluconazole, isavuconazole, itraconazole, hydroxy-itraconazole, posaconazole, and voriconazole [30]. Smith et al., proposed a method specifically for detecting voriconazole, isavuconazole, and posaconazole [31]. Yoon et al., reported a quantification method for voriconazole, itraconazole, and posaconazole [32]. Müller proposed a method for quantifying six agents, including fluconazole, isavuconazole, itraconazole, hydroxy-itraconazole, posaconazole, and voriconazole [33]. However, these methods detect only one class of agents (triazoles), whereas different classes of agents are commonly used in clinical combination therapy. For example, amphotericin B (a polyene) is used in combination with fluconazole (a triazole). Therefore, our method is suitable for rapid TDM of combination therapies, which is very useful in clinical practice. In terms of sample volume, most methods require 100 or 50 μL of plasma. Our assay requires 50 μL of serum, whereas an assay developed by Yoon et al., only required 10 μL. This reduction in sample volume may be useful in the case of pediatric and intensive care patients. However, in most cases, a serum volume of 50 μL or 100 μL is considered small and acceptable. In terms of sample preparation, all assays involve the use of methanol or acetonitrile for deproteinization owing to their simplicity and effectiveness. Chromatographic conditions vary among the methods mentioned above. The previously mentioned methods utilize ammonium acetate in the aqueous mobile phase for chromatographic separation; however, ammonium acetate was not added to our study, as its inclusion did not significantly enhance peak characteristics. Thus, compared with the methods described above, the mobile phase of our method is cheaper, and its preparation is faster. Additionally, the linear range of all agents in our method is consistent, simplifying the preparation of standard and quality control samples and reducing unnecessary errors.

## 3. Materials and Methods

### 3.1. Chemicals and Reagents

Voriconazole and VOR-IS were obtained from Sichuan Meidakang Huakang Pharmaceutical Co., Ltd. (Sichuan, China). Fluconazole was provided by the NPEL-TRACE Standard Technical Services Co., Ltd. (Shanghai, China). Posaconazole was purchased from CATO (Eugene, OR, USA). Isavuconazole and tylosin were purchased from Aladdin (Shanghai, China). Caspofungin was purchased from TRC (Toronto, ON, Canada). Micafungin was supplied by Hisun Pharmaceutical Co., Ltd. (Zhejiang, China). Amphotericin B was obtained from North China Pharmaceutical Group (Hebei, China). Ethylparaben was purchased from Sigma-Aldrich (St. Louis, MO, USA). Chromatography grade methanol, acetonitrile, and formic acid (FA) were purchased from Thermo Fisher Scientific (Waltham, MA, USA). Ultrapure water was obtained from an ultrapure water system (Zhiang, Shanghai, China). Dimethyl sulfoxide (DMSO) was purchased from Meryer Co., Ltd. (Shanghai, China).

### 3.2. Serum Sample Collection

Venous blood samples from healthy donors and patients were collected at the Zhongnan Hospital of Wuhan University. Prior to blood collection, all participants were informed about the study and signed an informed consent form. This study was approved by the Medical Ethics Committee of the Zhongnan Hospital of Wuhan University (batch number: 2022238). The blood samples were collected in a yellow tube containing a coagulant and separation glue, followed by immediate centrifugation (3000× *g*, 10 min at 4 °C). The resulting serum was carefully collected and stored at −80 °C until further use.

### 3.3. Sample Preparation

First, the collected frozen serum samples were thawed at room temperature, and 50 µL of each sample was transferred into a fresh 1.5 mL polypropylene tube. Subsequently, 240 μL of methanol (containing 0.1% FA) and 10 μL of IS solution (40 μg/mL, comprising three mixtures of ISs) were added to the tube to facilitate protein precipitation. The mixture was thoroughly vortexed and centrifuged at 12,000× *g* for 10 min. The clear supernatant (250 μL) was collected and transferred to a new auto-sampler vial for testing.

### 3.4. Chromatographic and Mass Spectrometric Conditions

Sample analysis was performed using a triple quadrupole mass spectrometer interfaced with an electrospray ion source (Shimadzu 8050, Kyoto, Japan). The separation was carried out on a Waters Acquity UPLC C8 column (1.7 μm, 2.1 mm × 50 mm). The column oven temperature was kept at 40 °C. The injection volume was 2 μL. The mobile phase consisted of water with 0.1% FA (mobile phase A) and acetonitrile with 0.1% FA (mobile phase B). The flow rate was set at 0.4 mL/min. The gradient elution parameters were set as follows: 0 min-0.5 min-3 min-4 min-7 min-9 min-10 min, B 5%-5%-50%-100%-100%-5%-5%. The ESI parameters included a sheath gas flow rate of 3.0 L/min, an auxiliary gas flow rate of 10 L/min, an ion transfer tube temperature maintained at 300 °C, and a vaporizer temperature at 400 °C. Furthermore, the desolvation temperature was set at 250 °C, and the samples were measured in MRM mode with both positive and negative ionization modes.

### 3.5. Preparation of Stock Solutions, Calibrations, and Quality Controls

Based on their solubility, the pure analytes were dissolved in different solvents for preparing stock solutions (10 mg/mL). Posaconazole, voriconazole, isavuconazole, and amphotericin B were dissolved in DMSO, whereas caspofungin, micafungin, and fluconazole were dissolved in water. Subsequently, a mixed stock solution was prepared at 100 μg/mL concentration by mixing 10 μL of each stock solution with 930 μL of the blank serum. The stock solution of the IS (1 mg/mL) was stored in sterile containers at −80 °C until further use. Calibration standards were prepared through sequential dilution to obtain concentrations of 0.1, 0.2, 0.5, 2, 5, 10, and 50 µg/mL. Various quality controls (including LLOQ, LQC, MQC, and HQC, were prepared at concentrations of 0.1, 0.3, 5, and 40 μg/mL. All standard and QC samples were then partitioned into 0.5 mL aliquots and stored at −80 °C.

### 3.6. Method Validation

#### 3.6.1. Linearity

Calibration curves were generated for each analyte by plotting the analyte/IS peak area ratio against the nominal concentrations. For acceptance, the coefficient of variation in the LLOQ was required to be less than 20%, and the accuracy bias was expected to be within 20%.

#### 3.6.2. Accuracy and Precision

Assay precision and accuracy were estimated via QC. Within- and between-run analyses were performed in four independent runs with six replicates at four concentrations (LLOQ, LQC, MQC, and HQC). Precision, expressed as the coefficient of variation (%CV), was required to be less than 15% at the LQC, MQC, and HQC levels, and less than 20% at the LLOQ level.

#### 3.6.3. Selectivity

Six blank biological samples were collected from healthy volunteers to assess their selectivity. Additionally, two hemolytic, two lipemic, and two icteric serum samples were collected and pooled to create blank serum without any analytes of interest to evaluate potential interference. The selectivity assessment involved the analysis of blank samples spiked with the IS and the various sources of blank samples mentioned above.

#### 3.6.4. Matrix Effect and Extraction Recovery

The matrix effect was evaluated by comparing the peak areas of the analytes spiked into the extracted biological matrices with those of pure solutions of the same concentration. The experiments were conducted according to the experimental scheme proposed by Matuszewski et al. [34]. The relative standard deviation of the normalized factor was required to be less than 20% for acceptance. After extraction, the analyte levels in the spiked serum samples were compared with those in the non-spiked samples. The extraction recoveries were determined at the LQC and HQC levels by comparing the peak areas of the serum samples spiked with all seven analytes. Three technical replicates were analyzed to evaluate the extraction recovery.

#### 3.6.5. Stability

The stability of the samples was evaluated by analyzing the LQC and HQC levels of the samples in terms of short-term, freeze-thaw, and long-term stability. QC samples were subjected to different storage conditions for assessment, including 48 h of incubation at 4 °C for short-term stability, four freeze–thaw cycles for freeze–thaw stability, and six months of incubation at −80 °C for long-term stability. For evaluating the freeze–thaw stability, in each cycle, the samples were frozen at −80 °C for a minimum of 12 h before being thawed. Freshly prepared QC samples were used as controls. According to the FDA guidelines, changes in concentration within 15% were acceptable for stability assessments [35].

### 3.7. Statistical Analysis

Data management and statistical analyses were performed using Microsoft Excel 2020. All statistical analyses were conducted using IBM SPSS software (version 22.0, IBM Corp., Armonk, NY, USA).

## 4. Conclusions

This study describes the development and validation of an LC–MS/MS-based method for the simultaneous quantification of seven antifungal agents in human serum. Only 50 μL of serum is required for analysis, and the sample preparation process involves straightforward protein precipitation using methanol. The effectiveness of this method in producing accurate and precise results was verified, and the method was successfully used to measure the concentrations of antifungal agents in sera. The approach offers a more efficient and streamlined process for analyzing the concentrations of antifungal agents, particularly for therapeutic drug monitoring of antifungals used in combination therapy.

## Figures and Tables

**Figure 1 pharmaceuticals-16-01537-f001:**
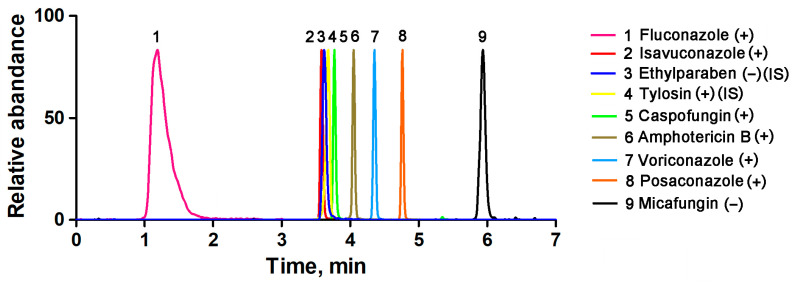
Typical chromatograms of the seven antifungal agents in human serum samples.

**Figure 2 pharmaceuticals-16-01537-f002:**
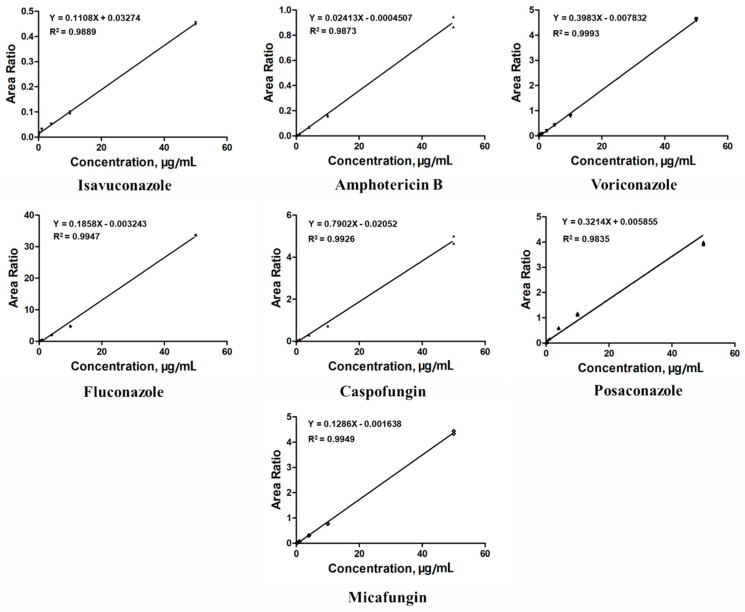
Calibration curves of the seven antifungal agents. The least-squares regression method was used with a linear regression weighting factor of 1/x^2^. All correlation coefficients (R^2^) are at least 0.9835 or better.

**Figure 3 pharmaceuticals-16-01537-f003:**
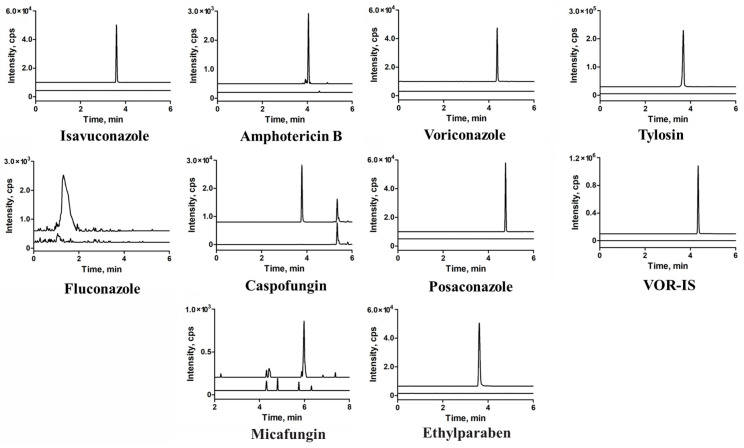
Overlapping of the multiple reaction monitoring (MRM) chromatograms obtained from blank human serum and lower limits of quantification (LLOQs) of the seven antifungal agents, blank versus blank plus each internal standard were also present.

**Figure 4 pharmaceuticals-16-01537-f004:**
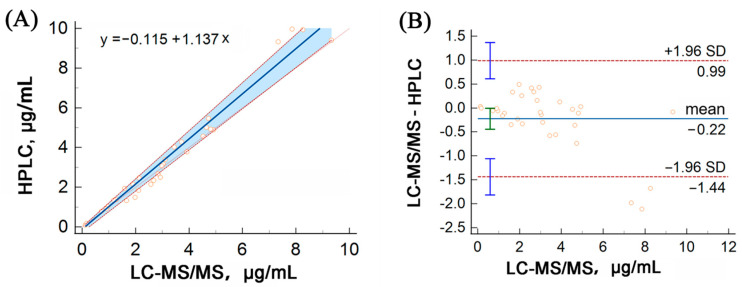
Method comparison of our LC–MS/MS assay versus HPLC for the detection of voriconazole using Passing–Bablok (**A**) and Bland–Altman (**B**) analyses. The solid line represents the regression line, while the dashed lines represent the 95% confidence interval for the regression line.

**Figure 5 pharmaceuticals-16-01537-f005:**
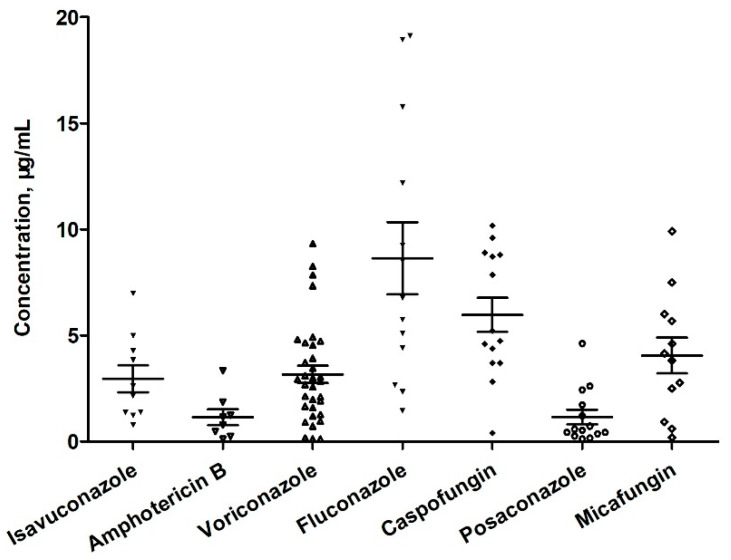
Results of the measurement of antifungal agents in serum samples collected from patients.

**Table 1 pharmaceuticals-16-01537-t001:** Mass spectrometry parameters for the detection of the seven antifungal agents. Micafungin and ethylparaben were detected in ESI (−) mode.

Compound	Retention Time (min)	Molecular Weight	Precursor Ion (*m*/*z*)	Product Ion (*m*/*z*)	Collision Energy (eV)	Dwell Time (msec)	Linear Range (μg/mL)	IS
**Isavuconazole**	3.60	716.74	717.20	236.15	−10	20	0.1–50	VOR-IS
				165.10	−10	20		
**Amphotericin B**	4.05	924.08	906.50	743.50	−10	20	0.1–50	VOR-IS
				725.40	−10	20		
**Voriconazole**	4.37	349.31	350.20	281.20	−5	20	0.1–50	VOR-IS
				127.20	−5	20		
**Fluconazole**	1.29	306.27	307.10	220.10	−19	20	0.1–50	Tylosin
				238.10	−5	20		
**Caspofungin**	3.78	1093.31	547.40	137.10	−30	20	0.1–50	Tylosin
				131.10	−26	20		
**Posaconazole**	4.76	700.78	701.30	614.30	−36	20	0.1–50	VOR-IS
				344.20	−47	20		
**Micafungin**	6.02	1270.27	1268.70	246.95	50	20	0.1–50	Ethylparaben
				319.91	65	20		
**VOR-IS**	4.37	354.31	355.3	284.10	−5	20		
				128.20	−5	20		
**Tylosin**	3.74	916.10	916.75	772.50	−35	20		
				174.20	−30	20		
**Ethylparaben**	3.67	166.18	165.15	92.05	35	20		
				136.70	20	20		

**Table 2 pharmaceuticals-16-01537-t002:** Intra-day and inter-day accuracy and precision data for assay validation.

**Compound**	**LLOQ**					**LQC**				
	**Concentraion** **(μg/mL)**	**Mean** **(μg/mL)**	**%Bias**	**Intra-Day Precision (%CV)**	**Inter-Day Precision (%CV)**	**Concentraion** **(μg/mL)**	**Mean** **(μg/mL)**	**%Bias**	**Intra-Day Precision (%CV)**	**Inter-Day Precision (%CV)**
**Isavuconazole**	0.10	0.116	16.1	9.4	12.8	0.30	0.341	13.6	6.2	10.2
**Amphotericin B**	0.10	0.109	9.2	7.9	11.3	0.30	0.312	4.2	7.3	9.4
**Voriconazole**	0.10	0.105	4.6	4.8	8.4	0.30	0.332	10.7	4.1	9.3
**Fluconazole**	0.10	0.103	2.8	4.4	4.1	0.30	0.302	0.8	2.3	5.4
**Caspofungin**	0.10	0.104	4.3	8.9	12.7	0.30	0.332	10.8	9.2	11.4
**Posaconazole**	0.10	0.109	8.9	11.9	13.2	0.30	0.316	5.3	6.6	6.7
**Micafungin**	0.10	0.112	11.9	8.3	10.4	0.30	0.322	7.3	8.2	7.9
**Compound**	**MQC**					**HQC**				
	**Concentraion** **(μg/mL)**	**Mean** **(μg/mL)**	**%Bias**	**Intra-day Precision (%CV)**	**Inter-day Precision (%CV)**	**Concentraion** **(μg/mL)**	**Mean** **(μg/mL)**	**%Bias**	**Intra-day Precision (%CV)**	**Inter-day Precision (%CV)**
**Isavuconazole**	5.00	5.360	7.2	6.3	8.9	40.00	35.63	−10.9	8.2	13.2
**Amphotericin B**	5.00	5.212	4.2	3.9	8.6	40.00	43.34	8.4	7.5	6.9
**Voriconazole**	5.00	5.423	8.5	3.2	5.7	40.00	39.60	1.0	3.5	3.1
**Fluconazole**	5.00	5.072	1.4	1.2	3.2	40.00	44.34	10.9	2.1	2.4
**Caspofungin**	5.00	5.136	2.7	6.7	8.3	40.00	39.45	−1.4	11.2	9.3
**Posaconazole**	5.00	5.446	8.9	5.6	4.2	40.00	37.45	−6.4	10.3	12.5
**Micafungin**	5.00	5.352	7.0	8.2	7.6	40.00	42.52	6.3	9.3	13.2

**Table 3 pharmaceuticals-16-01537-t003:** Extraction recovery and matrix effect of the seven antifungal agents determined in LQC and HQC samples. RSD, relative standard deviation; LQC, low-quality control; HQC, high-quality control.

Compound	Recovery %				Matrix Effect			
	LQC	RSD	HQC	RSD	LQC	RSD	HQC	RSD
**Isavuconazole**	91.23	7.3	87.23	9.2	1.68	6.0	2.17	10.3
**Amphotericin B**	95.22	5.3	104.34	8.5	2.31	8.3	2.21	3.6
**Voriconazole**	94.33	4.7	105.24	4.1	1.68	6.2	1.81	5.8
**Fluconazole**	92.11	9.5	87.35	4.3	1.39	10.9	1.47	6.2
**Caspofungin**	94.23	8.7	90.27	11.7	2.14	12.5	1.94	7.2
**Posaconazole**	92.65	3.6	82.27	2.2	1.17	8.4	1.42	3.1
**Micafungin**	85.34	5.8	82.45	6.7	1.29	10.4	1.41	3.5

**Table 4 pharmaceuticals-16-01537-t004:** Stability of quality control (QC) samples under different conditions, data are represented as mean (percentage).

Compound	Concentration (μg/mL)		Freshly Prepared QC (μg/mL)		4 °C for 48 h (μg/mL)		Four Freeze-Thraw Cycles (μg/mL)		−80 °C for 3 Months (μg/mL)	
	LQC	HQC	LQC	HQC	LQC	HQC	LQC	HQC	LQC	HQC
**Isavuconazole**	0.30	40.00	0.332 (106.7)	36.89 (92.2)	0.322 (107.3)	36.24 (90.6)	0.293 (97.7)	35.45 (88.63)	0.314 (104.7)	40.83 (102.1)
**Amphotericin B**	0.30	40.00	0.313 (104.3)	42.12 (105.3)	0.304 (101.3)	42.13 (105.8)	0.286 (95.3)	41.21 (103.0)	0.313 (104.3)	42.22 (105.6)
**Voriconazole**	0.30	40.00	0.327 (109.1)	40.23 (100.6)	0.314 (104.7)	41.23 (103.1)	0.307 (102.3)	40.27 (100.7)	0.318 (106.0)	40.31 (100.7)
**Fluconazole**	0.30	40.00	0.296 (98.7)	43.21 (108.0)	0.283 (94.3)	42.12 (105.3)	0.288 (96.0)	41.29 (103.2)	0.297 (99.0)	42.97 (107.4)
**Caspofungin**	0.30	40.00	0.331 (110.3)	39.23 (98.1)	0.333 (111.0)	40.23 (100.6)	0.314 (104.7)	39.56 (98.9)	0.323 (107.7)	39.49 (98.7)
**Posaconazole**	0.30	40.00	0.314 (104.6)	38.22 (95.6)	0.311 (103.7)	38.11 (95.28)	0.302 (106.7)	37.11 (92.8)	0.315 (105.0)	38.33 (95.83)
**Micafungin**	0.30	40.00	0.318 (106.2)	41.21 (103.0)	0.312 (104.0)	40.38 (101.0)	0.307 (102.3)	39.29 (98.2)	0.312 (104.0)	40.86 (102.2)

**Table 5 pharmaceuticals-16-01537-t005:** Comparison of different methods for detection of multiplex antifungal agents using LC-MS/MS. The parameters include the compounds detected, lower limit of quantification (LLOQ), sample volume, and run time.

Compound	LLOQ (μg/mL)	Linear Range (μg/mL)	Sample Volume (μL)	Run Time (min)	Reference
Voriconazole	0.01	0.01–20	100	3.0	30
Posaconazole	0.02	0.02–40			
Fluconazole	0.2	0.2–200			
Itraconazole	0.02	0.02–20			
Hydroxy-itraconazole	0.01	0.01–10			
Voriconazole		0.5–10	50	6.0	31
Isavuconazole	0.1	0.5–10			
Posaconazole		0.17–8			
Voriconazole	0.1	0.1–30	10	3.8	32
Itraconazole	0.05	0.05–10			
4OH-itraconazole	0.05	0.05–10			
Posaconazole	0.05	0.05–10			
Fuconazole	0.0283	0.5–40	50	3.0	33
Isavuconazole	0.001	0.1–9			
Itraconazole	0.0017	0.1–4			
OH-ITZ	0.0262	0.05–4			
Posaconazole	0.103	0.05–8			
Voriconazole	0.006	0.1–6			
Isavuconazole	0.1	0.1–50	50	10.0	our assay
Amphotericin B					
Voriconazole					
Fluconazole					
Caspofungin					
Posaconazole					
Micafungin					

## Data Availability

Data is contained within the article.

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
