# Peer review of "Simultaneous Quantification of Seven Antifungal Agents in Human Serum Using Liquid Chromatography-Tandem Mass Spectrometry"

_pharmaceuticals, 2023, doi:10.3390/ph16111537_

Round 1

Reviewer 1 Report

Comments and Suggestions for Authors

The manuscript entitles ‘Simultaneous quantitation of seven antifungals in human serum using liquid chromatography-tandem mass spectrometry’ propose a new LC-MS/MS method for monitoring seven antifungal agents in human serum. I propose the publishing of the paper, after minor revision.

Row 74: ultraviolet detection (HPLC-UV)

Row 76: cannot analyze

Row 123: Phase B was not described in the text

Row 126-127: 5%, 50%... of what?

Row 129: For a visual representation, a representative chromatogram is given in Figure 1.

Row 131: Figure 1: If you put (-) sigh for some compounds, please put the appropriate sigh for the others too!

Table 2: Some RSD values obtained for inter-day precision are higher than the corresponding ones for the intra-day precisions, which is not analytically correct. Moreover, you have more RSD values higher for the higher concentration domains than for the lower ones, which couldn't be possible. The higher concentration domain is, the lower RSD will be. Please reevaluate all RSD values!

Table 4: Please express the values ± RSD

Row 253: Please use the same measurement unit all over the text

Author Response

The manuscript entitles ‘Simultaneous quantitation of seven antifungals in human serum using liquid chromatography-tandem mass spectrometry’ propose a new LC-MS/MS method for monitoring seven antifungal agents in human serum. I propose the publishing of the paper, after minor revision.

Thank you for your encouraging comments. We have revised the manuscript according to your comments.

Row 74: ultraviolet detection (HPLC-UV)

We have done according to your comments.

Row 76: cannot analyze

We have revised the manuscript according to your comments.

Row 123: Phase B was not described in the text

     Thank you for your reminder, the composition of the B phase is described in the Materials and Methods section.

Row 126-127: 5%, 50%... of what?

The sentence was reworded as “After testing different gradients, finally, the gradient was set as 5% (0 min), 50% (3 min), 100% (4 min), 100% (7 min), 5% (9 min), and 5% (10 min), all the gradients mentioned refer to phase B.”

Row 129: For a visual representation, a representative chromatogram is given in Figure 1.

We have revised the manuscript according to your comments.

Row 131: Figure 1: If you put (-) sigh for some compounds, please put the appropriate sigh for the others too!

We have revised the figure according to your comments. As shown below,

Table 2: Some RSD values obtained for inter-day precision are higher than the corresponding ones for the intra-day precisions, which is not analytically correct.

Thank you for your comments. Intra-day precision refers to the precision measured within the same day, using the same batch of samples, reagents, and standard curve. It evaluates the repeatability of the method. Inter-day precision refers to the precision of the analysis method when the determination of results is conducted at different times. It aims to investigate the performance of the instrument, reagents, standard curve, environmental conditions, and other minor changes that may occur during the sample determination process, leading to variations in the determination results.

As far as we know, there is no direct relationship between their values. Specifically, it should be noted that intra-day precision only represents data from a single day. The value may also vary from day to day.

We have also consulted relevant literature [1-4], which confirms that there is no numerical size relationship between these values.

Ref:

[1] Lu W, Pan M, Ke H, Liang J, Liang W, Yu P, Zhang P, Wang Q. An LC-MS/MS method for the simultaneous determination of 18 antibacterial drugs in human plasma and its application in therapeutic drug monitoring. Front Pharmacol. 2022, 13:1044234.

[2] Oswald S, Peters J, Venner M, Siegmund W. LC-MS/MS method for the simultaneous determination of clarithromycin, rifampicin and their main metabolites in horse plasma, epithelial lining fluid and broncho-alveolar cells. J Pharm Biomed Anal. 2011, 55(1):194-201.

[3] Ohmori T, Suzuki A, Niwa T, Ushikoshi H, Shirai K, Yoshida S, Ogura S, Itoh Y. Simultaneous determination of eight β-lactam antibiotics in human serum by liquid chromatography-tandem mass spectrometry. J Chromatogr B Analyt Technol Biomed Life Sci. 2011, 879(15-16):1038-42. 

[4] Jourdil JF, Tonini J, Stanke-Labesque F. Simultaneous quantitation of azole antifungals, antibiotics, imatinib, and raltegravir in human plasma by two-dimensional high-performance liquid chromatography-tandem mass spectrometry. J Chromatogr B Analyt Technol Biomed Life Sci. 2013, 919-920:1-9. 

Moreover, you have more RSD values higher for the higher concentration domains than for the lower ones, which couldn't be possible. The higher concentration domain is, the lower RSD will be. Please reevaluate all RSD values!

Thank you for your comments. This is indeed an interesting issue, as it pertains to a statistical problem. In general, when using the least-squares regression method with linear regression, higher concentration samples should exhibit lower RSD values. This is because standards with higher concentrations contribute to a larger proportion of the scale, resulting in smaller calculation errors. However, in our study, standard curves were constructed using a weighting factor of 1/x2, rather than the commonly used weighting factor of 1. This adjustment of weights for lower concentration standards eliminates the phenomenon.

For example, in the article by Mark E. Arnold et al.[1], they utilized different weighting factors to construct standard curves. As shown in their article Table 3b or Table 3c, the RSD data varied significantly depending on the choice of weighting factor. Furthermore, there was no correlation observed between changes in concentration and the RSD value.

Ref :

[1] Gu H, Liu G, Wang J, Aubry AF, Arnold ME. Selecting the correct weighting factors for linear and quadratic calibration curves with least-squares regression algorithm in bioanalytical LC-MS/MS assays and impacts of using incorrect weighting factors on curve stability, data quality, and assay performance. Anal Chem. 2014, 86(18):8959-66.

Table 4: Please express the values ± RSD

Thank you for your suggestion, in our opinion, our data expressed as value (percentage) can more intuitively reflect the differences between measured values and theoretical value, so we keep the expression unchanged.

Row 253: Please use the same measurement unit all over the text

We have revised the sentence as “Blood concentration monitoring was conducted, revealing a voriconazole concentration of 9.96 μg/mL,”

Reviewer 2 Report

Comments and Suggestions for Authors

Dear the Editor

Li W et al reported a method for simultaneous determination of seven antifungals by LC-MS/MS. Linearity was examined between 0.1-50 ug/mL. In Fig. 5, these authors showed the concentration of these antifungals in patients, presenting robustness of this assay for clinical specimen. Under this chromatographic condition, six analytes migrated with a sharp peak while fluconazole migrated at 1-2 min with a broad peak. Overall, the validation study has been performed properly. This is a well-organized paper providing sufficient data for TDM application.

Major concerns:

1) None.

Minor concerns:

1) Size of characters should be consistent thoroughly manuscript (LL13-14, LL317-323).

2) Unnecessary capitalization needs to be corrected (L63, L73).

3) In L145, VOR-IS and ETH need to be explained.

4) In 2.2.4 (LL204-208), sample storage condition needs to be described.

5) In 4.4 (L342-353), applied voltages and dwell time needs to be added for mass spectrometric condition.

Author Response

Li W et al reported a method for simultaneous determination of seven antifungals by LC-MS/MS. Linearity was examined between 0.1-50 ug/mL. In Fig. 5, these authors showed the concentration of these antifungals in patients, presenting robustness of this assay for clinical specimen. Under this chromatographic condition, six analytes migrated with a sharp peak while fluconazole migrated at 1-2 min with a broad peak. Overall, the validation study has been performed properly. This is a well-organized paper providing sufficient data for TDM application.

Thank you for your encouraging comments. We have revised the manuscript according to your comments.

Minor concerns:

(1) Size of characters should be consistent thoroughly manuscript (LL13-14, LL317-323).

These problems may have occurred in the layout process. We have revised the manuscript according to your comments.

(2) Unnecessary capitalization needs to be corrected (L63, L73).

We have revised the words according to your comments.

(3) In L145, VOR-IS and ETH need to be explained.

We have revised the words “VOR-IS ([13C2,2H3]-voriconazole), tylosin, and ethylparaben were selected as the ISs”

(4) In 2.2.4 (LL204-208), sample storage condition needs to be described.

   Thank you for your suggestion, the sample storage condition was described in the section 4.6.5, for more clearly, “For freeze-thaw stability, at each cycle, the samples were frozen at -80°C for a minimum of 12 hours before being thawed.” was added to this section.

(5) In 4.4 (L342-353), applied voltages and dwell time needs to be added for mass spectrometric condition.

     Yes, applied voltages and dwell time are important factor that need to be described in this study, so we added it in the Table 1.

Reviewer 3 Report

Comments and Suggestions for Authors

The study is more of a routine LC-MS analysis rather than a novel method development. The reported LC-MS method does not offer anything new or address any existing challenging issues regarding the determination of antifungal drugs in blood. 

Specific:

1. Table 1. The authors only used one MRM transition for each analyte, a practice which tends to lead to false positives. Suggest that the authors use at least two MRM transitions. This way relative ion ratio could be used to ensure better confidence. Additionally, the authors should have specified the identification criteria used for the study.

2. What's the dilution factor of the sample preparation? 6x? If so how could the authors quantitate the samples spiked at 0.3 ug/mL? The lowest calibration point is 0.1 ug/mL.

3. Section 4.4.: Clarify whether negative ionization mode was used.  

4. Table 1. What's the dwell time used for each MRM transition.

5. Line 113. The authors need to define "most effective separation". The peak shape of fluconazole looks terrible. Explain what could cause the tailing (Figure 1). Also there is no separation between peaks #2, 3, and 4. How could the authors justify this is an effective separation?

Comments on the Quality of English Language

 Minor editing of English language required.

Author Response

The study is more of a routine LC-MS analysis rather than a novel method development. The reported LC-MS method does not offer anything new or address any existing challenging issues regarding the determination of antifungal drugs in blood. 

   Thank you for your pertinent evaluation. This study is mainly about the application of mass spectrometry in TDM, rather than a methodological breakthrough. In our opinion, this study has very important application value for clinical practice, and it is appropriate for the Journal Issue of “Applications of Liquid Chromatography Coupled with Mass Spectrometry (LC-MS/MS) in Drug Analysis”

Specific:

(1) Table 1. The authors only used one MRM transition for each analyte, a practice which tends to lead to false positives. Suggest that the authors use at least two MRM transitions. This way relative ion ratio could be used to ensure better confidence. Additionally, the authors should have specified the identification criteria used for the study.

Thank you for your comments, in practice, we used two MRM transitions for detection, one for quantification, one for confirmation. And Table 1 was revised.

(2) What's the dilution factor of the sample preparation? 6x? If so how could the authors quantitate the samples spiked at 0.3 ug/mL? The lowest calibration point is 0.1 ug/mL.

Thank you for your comments. According to the Food and Drug Administration (FDA) Bioanalytical Method Validation Guidance for Industry, low-quality control is defined as three times the lower limit of quantification (LLOQ). In our study, the LLOQ is 0.1 μg/mL. Therefore, the low-quality control concentration is 0.3 μg/mL. It is important to note that these preparation processes do not require constant dilution factors.

(3) Section 4.4.: Clarify whether negative ionization mode was used.  

Thank you for your comments. A sentence “The electrospray ionization (ESI) source was operated in both positive and negative ionization modes in this study” has been added to the section.

(4) Table 1. What’s the dwell time used for each MRM transition.

Thank you for your comments. The dwell time was added to Table 1 in this revision.

  1. Line 113. The authors need to define "most effective separation". The peak shape of fluconazole looks terrible. Explain what could cause the tailing (Figure 1). Also there is no separation between peaks #2, 3, and 4. How could the authors justify this is an effective separation?

Thank you for your comments. As pointed out by the reviewer, in Figure 1, it is evident that several peaks are closely spaced. However, it is important to note that these peaks are monitored in different transitions, and their elution does not interfere with the quantitative analysis of each other.

We have added sentences “From a chromatographic perspective, the separation of the analytes was suboptimal. However, this was overcome by the use of compound-specific tandem mass spectrometric detection, which aided in their identification and quantification. Additionally, no interferences between the analytes were observed.” for clarification.

The presence of an undesired tailing in the peak shape of fluconazole may be attributed to the extraction solvent being considerably stronger than the mobile phase. We have experimented with different gradients, and under the condition mentioned in the manuscript, the shape of peak does not impact the subsequent quantitative analysis and can be deemed acceptable.

“The presence of an undesired tailing in the peak shape of fluconazole may be attributed to the extraction solvent being considerably stronger than the mobile phase.” have added to the manuscript.

Round 2

Reviewer 3 Report

Comments and Suggestions for Authors

In the revised version, the authors have not signified the novelty of the study, so I cannot recommend the current version for publications. There are already many published studies that focus on LC-MS determination of anti-fungal drugs in human blood. This study does not address any existing challenges or offer any additional benefits compared to published methods.

 Ref.

 Alffenaar et al. J Chromatogr B Analyt Technol Biomed Life Sci 2010, 878, 38-44.

 Toussaint et al., J Chromatogr B Analyt Technol Biomed Life Sci. 2017 1046, 26-33.

 McShane & Wang. Clin Chim Acta. 2017. 474, 8-13.

 Zheng & Wang. Clin Chim Acta. 2019, 491, 132-145.

Comments on the Quality of English Language

Minor editing of English language required.

Author Response

Thank you for your comments, and we have carefully reviewed the references you provided. In the case of hospitalized patients, antifungal drugs are commonly categorized into three groups: polyenes, triazoles, and echinocandins. However, the existing methods typically only detect one class of agents, specifically triazoles. Considering the significant morbidity and mortality associated with invasive fungal infections, combination therapy (different class of drugs) involving multiple antifungal agents may be an option, particularly in cases of severe infection.  For example, amphotericin B (a polyene) is used in combination with fluconazole (a triazole). Our method can simultaneously measure all three classes of drugs, polyenes (amphotericin B), triazoles (fluconazole, voriconazole, posaconazole, and isavuconazole), and echinocandins (caspofungin and micafungin). Therefore, our method offers more greater practicality for clinical applications, particularly for combination therapy. Additionally, the linear range of all drugs in our method is consistent, simplifying the preparation of standard and quality control samples and reducing the unnecessary errors.

Round 3

Reviewer 3 Report

Comments and Suggestions for Authors

The authors have only made some cosmetic changes in the revised version and the keys issues related to the significance of the study and insufficient method performance have not been fully addressed so I cannot recommend the manuscript for publication.

Comments on the Quality of English Language

The authors have only made some cosmetic changes in the revised version and the keys issues related to the significance of the study and insufficient method performance have not been fully addressed so I cannot recommend the manuscript for publication.